# Maxicircle architecture and evolutionary insights into *Trypanosoma cruzi* complex

**Luisa Berná**[1,2], **Gonzalo Greif**[1], **Sebastián Pita**[1,3], **Paula Faral-Tello**[1], **Florencia Díaz-Viraqué**[1], **Rita De Cássia Moreira De Souza**[4], **Gustavo Adolfo Vallejo**[5], **Fernando Alvarez-Valin**[2], **Carlos Robello**[1,6]*

**1** Laboratorio de Interacciones Hospedero-Patógeno, Unidad de Biología Molecular, Institut Pasteur de Montevideo, Montevideo, Uruguay, **2** Sección Biomatemática—Unidad de Genómica Evolutiva, Facultad de Ciencias, Universidad de la República, Montevideo, Uruguay, **3** Sección Genética, Facultad de Ciencias, Universidad de la República, Montevideo, Uruguay, **4** Grupo Triatomíneos, Instituto René Rachou, Fundação Oswaldo Cruz–Fiocruz, Belo Horizonte, Minas Gerais, Brazil, **5** Laboratorio de investigaciones en Parasitología Tropical (LIPT), Facultad de Ciencias, Universidad del Tolima, Tolima, Colombia, **6** Departamento de Bioquímica, Facultad de Medicina, Universidad de la República, Montevideo, Uruguay

* robello@pasteur.edu.uy

**Data Availability Statement:** The new maxicircle sequences were deposited in NCBI as with accesion numbers MW421590, MW421591, MW567142, MW567142, MW421592,

## Abstract

We sequenced maxicircles from *T. cruzi* strains representative of the species evolutionary diversity by using long-read sequencing, which allowed us to uncollapse their repetitive regions, finding that their real lengths range from 35 to 50 kb. *T. cruzi* maxicircles have a common architecture composed of four regions: coding region (CR), AT-rich region, short (SR) and long repeats (LR). Distribution of genes, both in order and in strand orientation are conserved, being the main differences the presence of deletions affecting genes coding for NADH dehydrogenase subunits, reinforcing biochemical findings that indicate that complex I is not functional in *T. cruzi*. Moreover, the presence of complete minicircles into maxicircles of some strains lead us to think about the origin of minicircles. Finally, a careful phylogenetic analysis was conducted using coding regions of maxicircles from up to 29 strains, and 1108 single copy nuclear genes from all of the DTUs, clearly establishing that taxonomically *T. cruzi* is a complex of species composed by group 1 that contains clades A (TcI), B (TcIII) and D (TcIV), and group 2 (1 and 2 do not coincide with groups I and II described decades ago) containing clade C (TcII), being all hybrid strains of the BC type. Three variants of maxicircles exist in *T. cruzi*: a, b and c, in correspondence with clades A, B, and C from mitochondrial phylogenies. While A and C carry maxicircles a and c respectively, both clades B and D carry b maxicircle variant; hybrid strains also carry the b- variant. We then propose a new nomenclature that is self-descriptive and makes use of both the phylogenetic relationships and the maxicircle variants present in *T. cruzi*.

## Introduction

As their name states, kinetoplastids are characterized by harboring the kinetoplast, a single branched mitochondria with an intricate organization of its own DNA, called kinetoplast DNA (kDNA). kDNA is a complex network of thousands of catenated circular DNAs of two

MW407947, and BioProject PRJNA713613 (https://www.ncbi.nlm.nih.gov/bioproject/PRJNA713613).

**Funding:** This work was supported by: Research Council United Kingdom Grand Challenges Research Funder under grant agreement 'A Global Network for Neglected Tropical Diseases' grant number MR/P027989/1 (CR grant); Agencia Nacional de Investigación e Innovación(UY) DCI-ALA/2011/023–502, 'Contrato de apoyo a las políticas de innovación y cohesión territorial', Fondo para la Convergencia Estructural del Mercado Común del Sur(FOCEM) 03/1 (CR grant); Fundacão de Amparo à Pesquisa do Estado de Minas Gerais/ CBB-AUC 00030-15 (RCMS grant). FDV has a doctoral fellowship from Agencia Nacional de Investigación e Innovación (ANII, Uruguay). LB, GG, SP, FDV, FAV and CR are members of the Sistema Nacional de Investigadores (ANII, Uruguay). The funders played any role in the study design, data collection and analysis, decision to publish, or preparation of the manuscript.

**Competing interests:** The authors have declared that no competing interests exist.

types, the minicircles and the maxicircles [1,2]. Maxicircles are equivalent to the mitochondrial genome of other eukaryotes, whereas minicircles are much shorter (seldom longer than 2 kb) and encode the guide RNAs (gRNAs) needed for editing most of the maxicircle transcripts [1,2]. During the editing, insertions and deletions of uridine residues at specific sites occur on target ARNs post-transcriptionally in order to obtain mature transcripts [3–5]. The complete genome of *Trypanosoma cruzi* maxicircles was first published in 2006 [6], in a comparative analyses of CLBrener and Esmeraldo strains, whose maxicircle sizes were estimated to be of 22Kb and 28Kb respectively, but a collapsed zone of repetitive sequences prevented their complete assembly. Recently, on a deep analysis of maxicircle divergent regions, Gerasimov et al. were able to "decompress" repetitive regions in trypanosomatids by using long reads [7]. Although similar at the taxonomic and structural level, trypanosomatids have very different lifestyles. In vertebrates, while *Leishmania spp.* infects cells belonging to the mononuclear phagocytic system, *T. brucei* remains extracellular, and *T. cruzi* is able to invade almost any kind of nucleated cells [8]. Their transmission vectors (phlebotomes, Tsetse flies and triatomines respectively) are evolutionarily very distant, so even intuitively one can anticipate that these parasites will use very different biological strategies to survive. In that context, species and genus are clearly defined in *Leishmania spp.*, for which a good correlation between species and clinical manifestations exists [9], and similarly, *T. brucei* taxonomy is well defined in those causing sleeping sickness in humans (*T. b. gambiense* and *T. b. rhodesiense*) [9]. However, the case of *T. cruzi* speciation remains still unclear. A few decades ago, two main clades of *T. cruzi* were described (I and II) based on biological and biochemical criteria, as well as molecular biology methods [10,11]. Subsequently, the use of sequences from genes and intergenic regions allowed the construction of gene phylogenies clearly showing that *T. cruzi* is composed of three major lineages which were called A, B and C, and that the distances between them were as large as that between *L. major* and *L. mexicana* [12]. This analysis introduced the concept of *T. cruzi* as a "species complex"instead of a single species. The three main clades were later confirmed by Machado and Ayala [13] that also described a fourth clade (D), and the presence of hybrid B/C strains. Afterwards, several analyses reinforced the view that the evolutionary relationships among *T. cruzi* "strains" cannot be reduced to a two groups scenario, proposing more complex relationships. This was accompanied by a change in the nomenclature, between letters and Roman numerals: the initial groups I and II were reclassified as I and IIa to IIe [14], and later on the six groups were numbered TcI to TcVI, and called "discrete typing units"(DTUs, [15,16]), where TcV and TcVI correspond to the hybrid lineages derived from haplotypes TcII (C) and TcIII (B). In addition it was postulated that bat-derived *T. cruzi* constitutes a seventh DTU [17]. Although in the last decade the "ABCD"denomination fell into oblivion, probably due to the high number of descriptive papers attempting to correlate DTUs to infected hosts, geographical areas, clinical manifestations, among others, the presence of three main evolutionary lineages was recently "rediscovered" in the form of a third musketeer [18], showing very similar results to those described ten years before [12,13].

In this work we obtained high quality assemblies of maxicircle genomes from the six DTUs of *T. cruzi*, including the resolution of repetitive regions, which allowed us in the first place to determine the precise architecture of maxicircles and its variations, and ensure a better knowledge and understanding of the evolution of *T. cruzi*.

## Results

### Complete maxicircle genomes of the six DTUs

For maxicircle genome analysis six strains were selected, one from each DTU: Dm28c (TcI), Y (TcII), MT3663 (TcIII), JoseJulio (TcIV), BolFc10A (TcV) and TCC (TcVI). The DTU

assignments of each of these strains were confirmed by multilocus PCR targeting the intergenic region of spliced leader genes (SL-IR), the 24Sα subunit ribosomal DNA (rDNA 24Sα) and the A10 fragments, as described by Burgos *et al.*[19] (S1 Fig). Using long reads from PacBio and Nanopore and post-corrected with Illumina reads, each maxicircle was assembled into a single circular contig, their sizes ranging from ~35Kb to ~50Kb (Table 1). Long read sequencing allowed us to determine that the maxicircles sequences of the six strains analyzed shared organization and compositional structure. We could identify four clearly defined regions conserved among them: the coding region (CR), two repetitive regions -the short (SR) and the long (LR) repeats- and an AT-rich region (<1kb) located between the coding region and the short repeat cluster (Figs 1 and 2A).

## Architecture of maxicircle genome

Maxicircle comparison clearly indicates that the four regions previously mentioned are conserved in the different lineages of *T. cruzi* (S2 Fig); however, whereas lengths of coding regions are relatively similar, significant differences were found among DTUs in the short (from ~2.1Kb to ~6.8Kb) and long (from ~14.3Kb to ~30.3Kb) repeats, as well as in the AT-rich region (from ~0,1Kb to ~1Kb), as summarized in Table 1. Nucleotide composition and skews clearly separate the four regions, and each one has a peculiar base composition (Figs 2A and S2). In the coding regions, nucleotide composition always correlates with gene orientation (+ or —strand) and the editing pattern (Figs 2A and S3); for example GC-skew (and conversely AT-skew) is lower in absolute number for the non-edited genes *nd2*, *nd1*, *coI*, *nd4* and *nd5*, changing from negative to positive depending on its orientation (*nd2*, *nd1*, *coI* in the negative strand; *nd4*, *nd5* in the positive strand; S3 Fig). Repetitive nature and different structure and composition of the SR and LR can be clearly visualized in the dotplots (Figs 3 and S4). On the one hand, despite the fact that the unit of repetition of the short repeat is not apparent, we found a consensus sequence of 67bp as a part of a longer repeat, with high AT content (77%) that is found in all but Y strain, with different levels of identity ranging from 75% to 100% (S1 Table). On the other hand, the long repeat also presents low identity among its monomers but notably, each one is delimited by a highly conserved element of 39 bp, which is present in all maxicircles in ~1–3 kb intervals depending on the strain, which is palindromic and consequently has the possibility to form cruciform structures (Fig 2C). By using maxicircles coding regions from *T. brucei*, *L. donovani*, *T. vivax* and *T. congolense* a similar pattern was found (S5 Fig).

## Insertions, deletions and gene truncations

Insertions and deletions were found in the maxicircle coding regions of BolFc10A and Y strains, respectively (Fig 2B and 2D). These variations are not due to artifacts in the assembly since many reads completely pass through insertions and deletions, with high coverage, ranging from 30x to 110x (S6A Fig). The Y strain (TcII) presents two deletions of 452

**Table 1. Statistics of assembled maxicircles.** Length of coding region, AT rich region, short and long repeat region are reported for each strain. (*) base pairs.

| Strain (DTU) | Length* (%GC) | Coding region* (%GC/%length) | Short repeat* (%GC/%length) | Long repeat* (%GC/%length) | AT rich region* (%GC) |
|---|---|---|---|---|---|
| Dm28c (TcI) | 50478 (24,1) | 15359 (25,4/30,4) | 4163 (21,1/8,2) | 30321 (23,9/60) | 635 (17,3) |
| Y (TcII) | 38789 (22,5) | 13852 (24,9/35,7) | 4913 (20,1/12,7) | 19887 (21,3/51.2) | 137 (23,9) |
| MT3663 (TcIII) | 44186 (25,7) | 15293 (26,2/34,6) | 5067(20,6/11,5) | 22830 (27,0/51.7) | 996 (17,1) |
| JoseJulio (TcIV) | 44279 (25,8) | 15263 (26,1/34,5) | 5196 (20,7/11,7) | 22878 (27,0/41.2) | 942 (17,4) |
| BolFc10A (TcV) | 34804 (26,8) | 17536 (27,7/50,4) | 2135 (21,2/6,1) | 14324 (27,1/41.2) | 809 (17,5) |
| TCC (TcVI) | 42479 (25,6) | 15339 (26,1/36,1) | 6797 (20,2/16) | 19343 (27,3/45) | 1000 (25,3) |

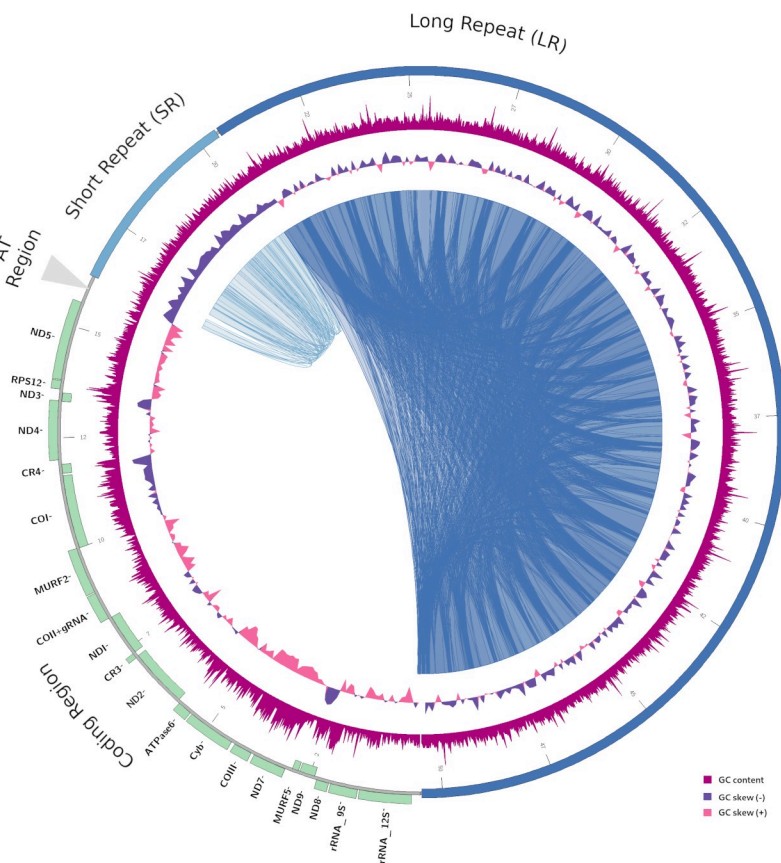

**Fig 1. Architecture of *T. cruzi* maxicircle.** Schematic representation of *Tc*Dm28c maxicircle. Repetitive regions are denoted by internal blast hits inside the circle (blue and light blue). The GC-skew (windows size 100 bp) are represented in violet (positive) and pink (negative). The GC content (window size 100 bp) is represented in magenta. The localization of all annotated genes are shown in the outer circle indicating their coding direction (outer + strand, inner—strand).

and 1071 bp, the first disrupting the *nd7* gene, and the second provoques the 5' deletion of *nd2* (110 bp), the complete elimination of *cr3*, and 3' deletion of *nd1* (780 bp), from which only remain the first 5' 167 nucleotides (Fig 2B). This insertion was not present in the TcII strains Berenice and Esmeraldo, however the same previously reported 236 bp deletion in Esmeraldo that disrupts *nd4* [6] is present in Berenice but not in Y (S7 Fig). On the other hand, BolFc10A strain (TcV) has two insertions of 1408 bp and 1017 bp length, separated by 893 bp (Fig 2B and 2D). Sequence analysis of both insertions shows that they belong to minicircle sequences, the first one of 1408 bp corresponds to an entire minicircle including the four conserved characteristic regions (Fig 2D), and the 1017 bp corresponds to a partial minicircle sequence, maintaining homology only at the conserved regions (Fig 2D). Although the insertions correspond to two different minicircles, in both cases they carry the same gRNA for *nd3* gene (S8 Fig). The minicircle conserved regions can be also visualized in the dotplots as a third repetitive region in TcV (Fig 3), and exhibit very high coverage of mapped reads, indicating that these sequences are probably present in the minicircle repertoire (S6B Fig) too. The first insertion disrupts the *nd4* gene (position 632), the second is located in the intergenic region between *nd4* and *nd3*, and a third one interrupts *nd2* (Fig 2B).

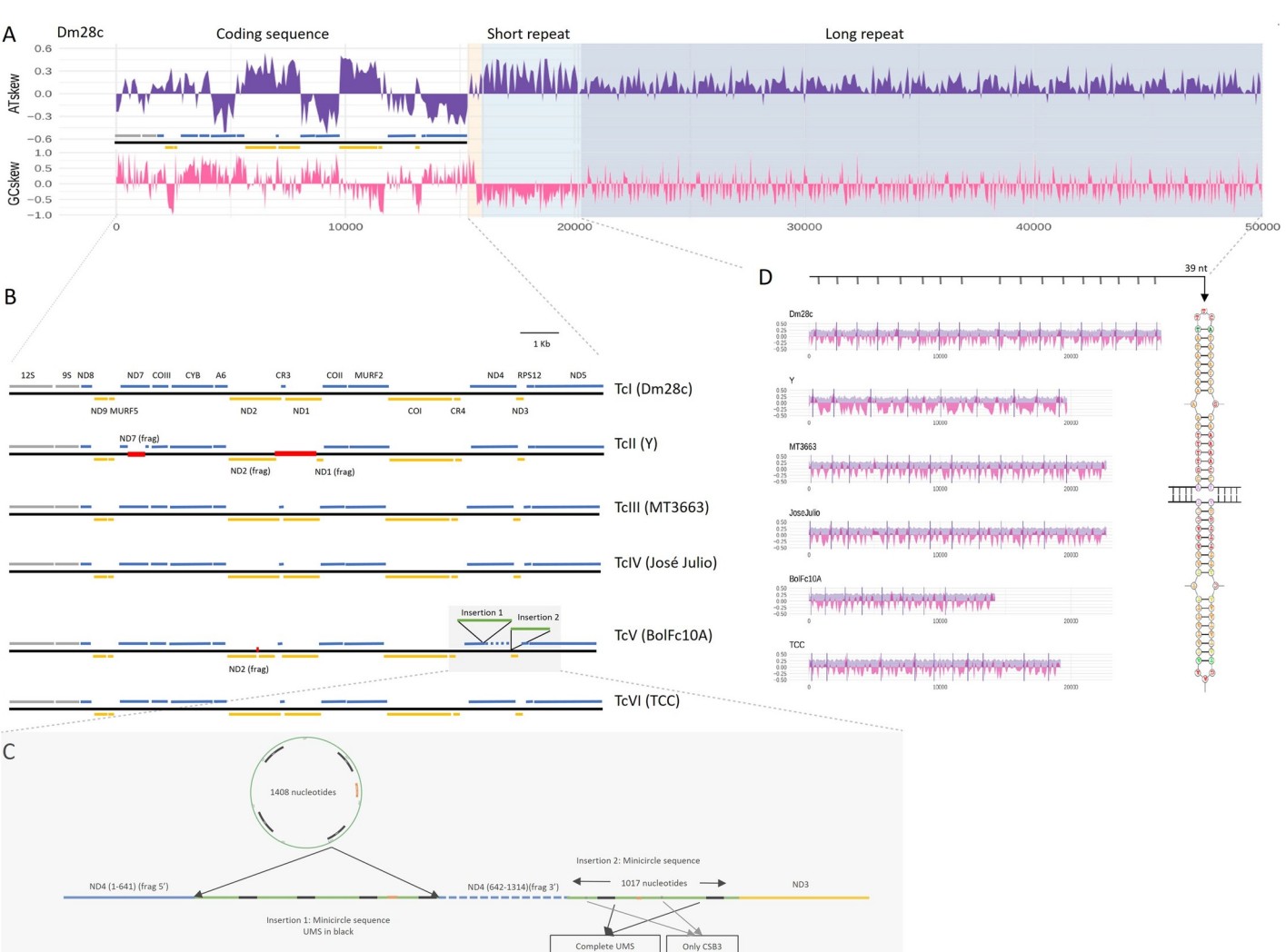

**Fig 2. Structure conservation among *T. cruzi* maxicircles. A.** Schematic representation of *Tc*Dm28c maxicircle including: coding region, AT-rich, short repeat region and long repeat region. AT-skew and GC-skew are represented in violet and pink respectively. **B.** Representation of coding region from the six DTU assemblies. Genes are indicated in blue (positive strand), orange (negative strand) or gray (ribosomal genes). Deletions are indicated with red lines, and insertions with green lines. **C.** Zoom of insertion in TcBolFc10A (see text). **D.** Conservation of structure of long repeat regions among DTU's showing the 39 bp palindromic sequence.

## Phylogenetic analysis of *T. cruzi* maxicircles

The phylogenetic analysis of the six DTUs by using the complete coding regions of maxicircles identifies three clearly delimited clades—A, B and C -, with an identical structure to that previously described [12], where A corresponds to TcI, B to TcIII-VI-V-VI, and C to TcII (Fig 4); B forms a compact group, whereas A and C present greater distances between them and to TcIII-TcVI with values of ~7 and ~10 respectively (Fig 4). In agreement with these data, similar results were recently observed [20,21]. We then sequenced (Illumina) more strains (S2 Table) with low coverage, but sufficient to obtain the entire maxicircle coding regions, and an identical clusterization structure was found (S9A Fig). In this last experiment three TcBat strains were included and it is clearly determined that they belong to clade A. From another perspective we can establish that in *T. cruzi* exist three maxicircles variants: a, b and c, and their differences not only depend on their coding regions: in the dotplots it can be visualized that they have differences in their patterns in the LR regions (Figs 3 and S4). It is worth

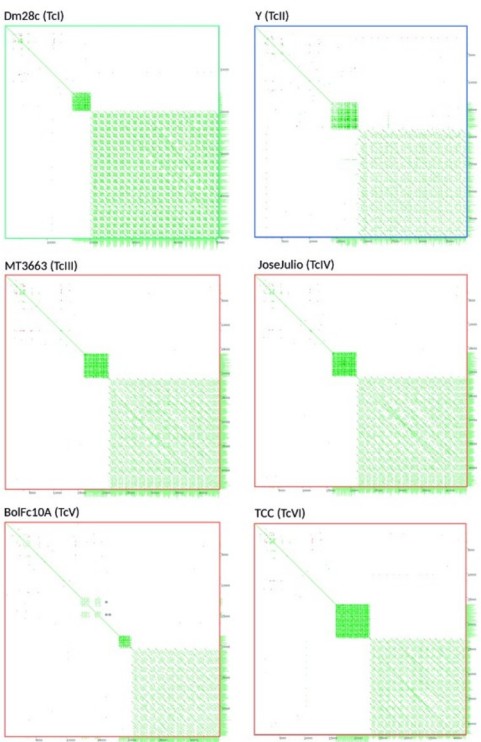

**Fig 3. Dotplot of maxicircles assemblies.** Dotplot visualizations by Yass [67] of self-self maxicircle of the six DTUs. Three main classes of maxicircle could be observed (green, blue and orange squares).

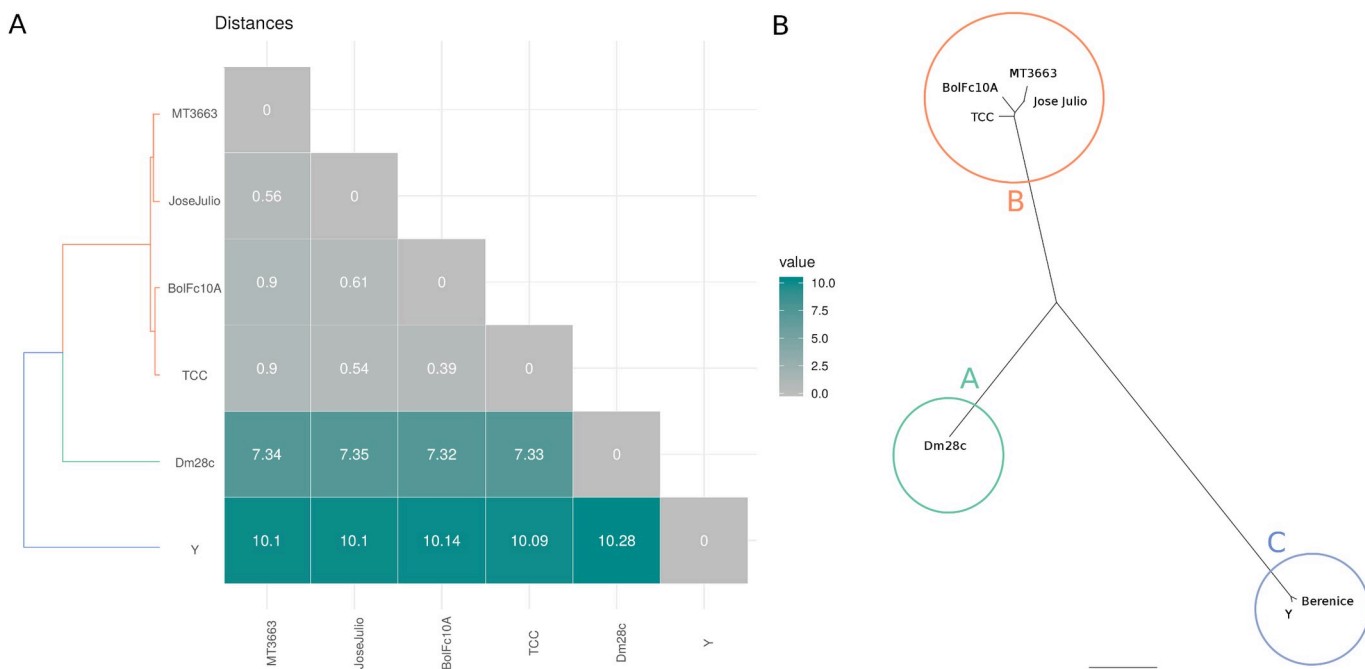

**Fig 4. Maxicircle phylogeny. A.** Matrix of all-against-all uncorrected p-distance. **B.** Phylogenetic maximum likelihood tree, unrooted visualization (Clades A, B and C are indicated by circles).

mentioning that the recently PacBio sequenced TcV strain Bug2148 [22], falls into clade A, closely related to Sylvio strain (S9B Fig). This unexpected phylogenetic location is also supported by a nuclear single-copy genes phylogeny (see S10 Fig and below) indicating that the strain named Bug2148 corresponds to TcI (probably Sylvio X10Cl1). This conclusion is reinforced by the fact that it carries the type "a" maxicircle, only present in TcI; that is why Bug2418 was not included in our analyzes. Recent phylogenetic analyses are in line with this observation [20,21].

## Mitochondrial *vs*. nuclear phylogenies

To get a whole picture of the evolutionary history and phylogenetic relationships of *T. cruzi*, a robust phylogeny was performed by identifying those nuclear genes having a unique copy in all of the DTUs. In addition to the strains used along this work, we included the available genomes from Sylvio (TcI) [23] CLBrener (TcVI/haplotypes TcII and TcIII) [24,25], 231 (TcIII) [26], AM64 (TcIV) [27], plus *T. cruzi marinkellei* as an outgroup, obtaining a list of 85 single-copy genes (S1 Data). It should be noted that although there are many unique genes, the different completeness of the genomes used results in the recovery of only a set of 85 unique conserved genes. The ML tree shown in Fig 5 (left), allows to identify two main clades, one early branching clade (Group 1) composed by C strains (TcII), and another (Group 2) composed by A, B and D strains (TcI, TcIII and TcIV respectively), being A, B, C and D all monophyletic. Every branching event is well supported by bootstrap values > 0.9. Moreover, when we excluded hybrid strains (TCC and TcBolFc10A), then 1108 (instead of 85) single-copy nuclear genes were identified (S2 Data), and the new phylogenetic ML tree obtained shows exactly the same topology with the four clades (S10 Fig). To compare nuclear and kDNA phylogenies, we used the same strains on the analyses with the addition of the already published Tc Esmeraldo strain kDNA sequence [6], to compare it with CLBrener-Esmeraldo like nuclear haplotype. The ML tree generated using all coding genes from the maxicircles is presented in Fig 5. Clades A and C show a clear correspondence between mitochondrial and nuclear clades, whereas the both B and D nuclear clades correspond with the mitochondrial clade B; every branching event is well supported by bootstrap values higher than 0.9.

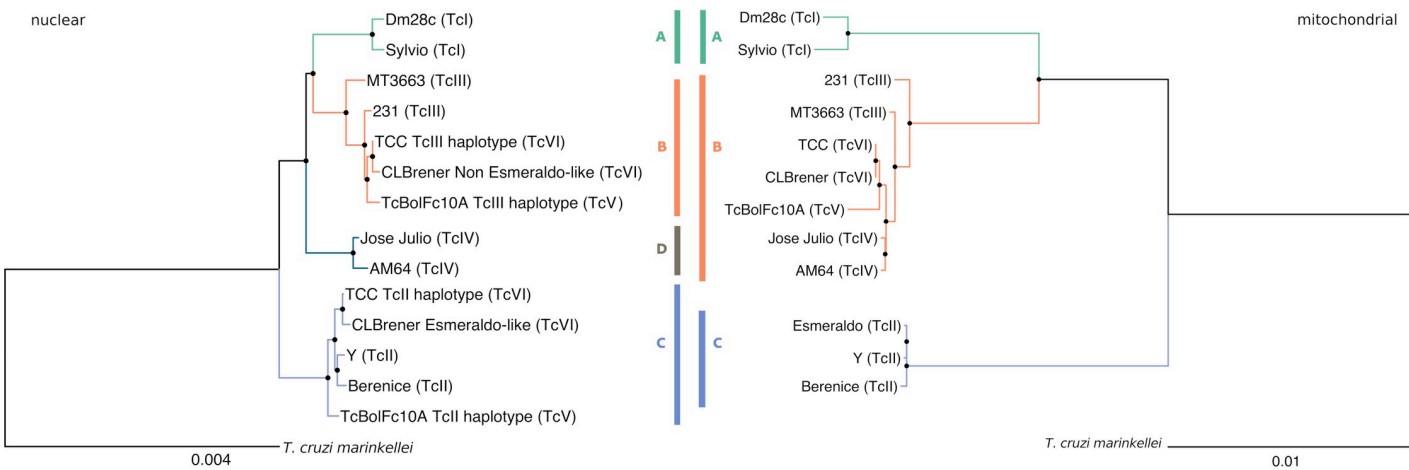

**Fig 5. Mitochondrial vs Nuclear phylogenies. A.** Nuclear phylogenetic maximum likelihood tree using 85 single-copy genes of 11 strains and *T. cruzi marinkellei* as outgroup. **B.** Mitochondrial (coding region) phylogenetic maximum likelihood tree corresponding to the same 12 strains. Nodes with bootstrap value higher than 0.9 are depicted with a black dot on the node.

## Discussion

The kinetoplast has been widely studied in trypanosomatids due to their distinctive properties, making it an attractive target for therapies for Chagas disease, leishmaniasis and sleeping sickness. It also constitutes a valuable phylogenetic marker for the reconstruction of trypanosomatids evolutionary history (reviewed by Kaufer *et al* [28]). In this work, by using a combination of short (Illumina) and long (PacBio and Nanopore) read DNA sequencing, a total of six *T. cruzi* maxicircles were sequenced and assembled. This strategy allowed us to determine their real length and structure. We found that in all cases their lengths were previously under-estimated, mainly due to the presence of two repetitive regions SR and LR (Table 1 and Fig 1), that collapsed during the assembly using first and second generation sequencing methods. The length of *T. cruzi* maxicircles ranges from 34,804 bp to 50,478 bp, which coincides with the variability described recently in other trypanosomatids, where also the main source of size differences is at expenses of repeats [7]. It is very important to highlight that these dimensions are compatible with reports from 3–4 decades ago. In an elegant work, Leon et al. [29] extracted kDNA from the Y strain, obtaining high degrees of purity from NaI gradients, and studied them by restriction patterns and electron microscopy, concluding that the approximate molecular weight was around 26x10$^6$ Da. If we consider that the molecular weight average of a single base pair is 650 Da, the deduced length from that publication is 40 kbp, and our results on Y strain agree with that pioneer work.

We found that the overall structure of *T. cruzi* maxicircles is conserved among DTUs, and consists in four regions: a) coding region (CR); b) AT-rich region (ATr); c) short repeat (SR); d) long repeat (LR), each one with a particular nucleotide composition (Table 1 and Figs 1 and 2). Regarding the AT-rich region (AT content 83%), its irruption indicates the end of the coding region, and its length ranges from 137 to 1000 bp. The changes in composition, as well as the AT-rich regions were associated with the mitochondrial replication origin [29,30]; in trypanosomatids, the origin of replication has been identified at different positions flanking the coding region either upstream (*T. brucei*) or downstream (*C. fasciculata*), but in both cases related to repetitive regions [31,32]. Concerning the short and long repeats, they exhibit vestigial monomers with low identity among them. Previously, similar structures have been described for *T. cruzi* as P5 and P12 elements, according to their proximity to the ND5 and 12Sgenes [7]. Our analysis revealed that the short repeats show similar composition among DTUs, and cover between 6.1% to 16% of maxicircle length (Table 1). A conserved region of 65 bp was identified in these repeats (S1 Table). Indeed, it was not possible to identify it on Y strain, although it is present in Esmeraldo and Berenice TcII strains. The long repeat covers, depending on the strain, between 41.2% to 60% of the total maxicircle length (Table 1). It does not show high sequence conservation among the different groups, but presents an inverted repeat composed by a conserved palindromic sequence of 39 bp (Fig 2C). This palindrome has been previously found in the first maxicircle genome reported, although only two copies were identified at that time [6]. Here we determined that it constitutes a hallmark of *T. cruzi* maxicircles, since it is present in all of them in at least eight copies defining the repetition unit of long repeats (Fig 2C). Palindromic structures were found in most mtDNAs studied [33] in chloroplasts and proteobacteria genomes [34,35], but their function is not known. In eukaryotes they have been associated with a diversity of functions like replication origins, and as targets for many architectural and regulatory proteins, such as histones H1 and H5, topoisomerase IIβ, HMG proteins, p53, among others [36]. Although the function of the 39 bp palindrome remains to be elucidated, its high degree of conservation and periodicity can be related to the ability of maxicircles to self-associate, even after elimination of RNA and proteins [37], and could be critical in kDNA structure. Taken together these observations lead us

to propose this new nomenclature (CR, ATr, SR and LR) to describe maxicircles architecture, instead of the current denomination of conserved and divergent regions (CR and DR). We recently found in *Trypanosoma vivax* the same short and long repeat structure, is present in both in American and African strains [38]. In addition, the dotplots obtained in this work for *T. brucei*, *L. donovani*, *T. vivax* and *T. congolense* (S5 Fig) strongly suggest this is a common pattern and hence, their functional roles deserve to be investigated.

The coding regions, reported to conserve the gene order among trypanosomatids, include genes encoding for members of the respiratory chain *nd* (subunits 1–5; 7–9), *co* (subunits I-III), *cyt b*, and *ATPase*, and the open reading frames of unknown function *murf* and *cr*. The *T. cruzi* complex not only conserves the order of genes among DTUs but also the strand location (Fig 2B). As was previously observed [6] GC or AT skews are good predictors of location of protein coding genes: positive GC-skew and AT-skew values represent genes in the plus and minus strand respectively, with the exception of *cr* genes (in agreement with their base composition:"c-rich genes"): *cr3* is surrounded by *nd2* and *nd3*, and no changes in AT skew are observed, and *cr4* exhibits the same pattern as *coI*, located in the opposite strand (Figs 2A and S3). Despite the high degree of conservation in the coding region, insertions and deletions were detected (Fig 2B). These variations may represent events that occurred exclusively in the particular sequenced strain or can they be common to a given lineage. Either way, this illustrates the degree of variability of *T. cruzi* maxicircles. Two deletions were found in Y (TcII), a 1071 bp deletion located in the intergenic region between *nd1* and *nd2*, with the consequent elimination of *cr3*, and a 452 bp deletion disrupting *nd7* (Fig 2B). It is worth noting that the same *nd7* truncation was already found in *T. cruzi* strains isolated from asymptomatic patients [39], where the authors analyze by PCR the *nd7* truncation, showing that it is not a feature of TcII. Although this deletion is not present neither in Esmeraldo nor in Berenice, we found that both strains exhibit a similar deletion affecting *nd4* gene. In addition, Berenice strain presents two further deletions affecting this gene (S7 Fig) similar to that described by Westenberger *et al.* [6]. In the case of TcV, BolFc10A presents an insertion that interrupts the *nd4* gene, whereas a second one falls on an intergenic region (Fig 2B). The finding of deletions always affecting mitochondrial genes encoding NADH dehydrogenase subunits, raises the question about the existence of a functional complex I in *T. cruzi* [40] in which, as with most eukaryotes, respiration occurs via the electron transport chain (ETC) coupled to ATP synthesis [41]. Decades ago it was clearly demonstrated that the main source of electrons in *T. cruzi* ETC is succinate instead of NADH: no inhibition of respiration was found after the addition of inhibitors of complex I, whereas both motility and respiration of epimastigotes were inhibited by malonate, a competitive inhibitor of the mitochondrial succinate dehydrogenase [42]. In view of our findings, it would be relevant to reevaluate cellular respiration in different strains with and without deletions of *nd* genes, to draw conclusions about the presence of a functional complex I in *T. cruzi*. In fact, a possibility is that the integrity and functionality of complex I would depend on the strain.

The two major insertions of TcV correspond either to a complete minicircle (1408 bp) or to an incomplete one (1017 bp), the former containing the four CSBs (Figs 2D and S8). It is remarkable that the presence of a complete -and even an incomplete- minicircle inserted in a maxicircle has not been reported before, and its presence could be a consequence of a horizontal transfer, from mini to maxicircles, since it has been documented the relatively high inter-minicircle recombination rate [43]. The presence of gRNA genes in maxicircles was reported in different trypanosomatids. In fact, initially it was postulated that gRNAs were encoded in maxicircles due to their presence in maxicircles of *L. tarentolae* [44], *C. fasciculata* [45], and *T. brucei*, where gMURFII-2 was found to be transcribed as an individual transcription unit from maxicircle [46], similar to what happens in minicircles. In the case reported here, also the

conserved regions are present, giving their gRNAs a genomic context for transcription; remarkably both sequences carry different gRNAs but directed to the same *nd3* gene (S8 Fig). The biological significance of these insertions is not clear, and at this point we are tempted to speculate with the possibility of the inverse flux: maxicircles can constitute seeds of minicircles, and what we "captured" was a snapshot of a dynamic process, which will probably end with the functional elimination of the "inserted" minicircles. In fact a "free" version of the larger mini-circle is also present, as indicated by the fact that sequencing depth is massively higher in the segment of the maxicicircle containing the insertion, shown in S6B Fig. In any case, there is still much to know about the origin and evolution of this fascinating process.

Three variants of maxicircles were detected: a, b and c, that correspond to the clades A, B, and C (Figs 4 and S9A). The non-hybrid TcI, TcII and TcIII-IV bear the maxicircles a, c and b respectively, whereas both the hybrid strains TcV and TcVI carry the b-maxicircle (lowercases are used to distinguish variants from clades). These three main clades exactly match with those previously proposed by us more than 20 years ago [12] and, since in that work nuclear sequences were used, the ABC clustering pattern would not be due to a bias for using maxicir-cle sequences. To evaluate this hypothesis a careful phylogenetic analysis was performed, using more than a thousand single-copy nuclear genes, confirming a correspondence between nuclear and mitochondrial trees (Fig 5), with the addition of a fourth D clade (Tc IV). This new clade is closely related to A and B clades, and carries the b-maxicircle. Clade D was ini-tially described by Ayala and Machado (2001) who, using the mitochondrial *CYb*, and the nuclear rRNA promoter genes, obtained three (A-C) or four (A-D) clades respectively, clade D corresponding to TcIV. It is worth wondering: what is the origin of clade D? Until now it has been a headache to place this clade (TcIV) in *T. cruzi* phylogenies. It is clear that A, B and D share a more recent common ancestor (compared to C), but why does D carry b-maxicircles? The explanation that B and D diverged from an ancestor already containing the b-maxicircle is highly unlikely, considering the results revealed by nuclear trees (Fig 5), where A, B and D are monophyletic, and two kind of maxicircles are present (a and b). A second hypothesis to account for this discrepancy is that mitochondrial transfer (introgression) occurred between B and D [13,47–50]. The proponents of this interpretation suggest that D lineage would have acted as the donor [48,49]. There are two points of concern about this introgression hypothe-sis. First, it appears unlikely the occurrence of an event involving exclusively mitochondrial "passage" without nuclear mixture and subsequent recombination. Although previous results are compatible with this view (reviewed in [49]), they are based on very limited datasets of nuclear genetic material. To tackle this contradictory situation only a deep comparative genome analysis between B and D genomes is necessary; if hybridization occurred between B and D lineages, vestigial mosaic genomes should be found. The second aspect where the results presented herein are at odds with previous proposals, involves the direction of the genetic transfer. In effect, the phylogenetic analysis presented here suggests that in the case of intro-gression, the direction was from B to D and not the other way around as suggested before. In our view, if D (TcIV, JoseJulio and AM64) was the donor, then its placement in the tree should be as the earliest branching group in the B clade of the mitochondrial tree (Figs 5 and S9).

At this point, it is necessary to revisit the classification and nomenclature of DTUs. Although it constitutes a very useful tool to genetically differentiate the members of the *T. cruzi* complex, we visualize two main concerns. First, revisiting the origin of nomenclature it can be observed that the initial classification into groups I and II does not correspond to the phylogeny of *T. cruzi* (B and D are separated from A). This can best be seen after the subdivi-sion into I and IIa-e, (which is at the basis of the current classification in DTUs I to VI respec-tively): A (Tc I) became separated from B and D. Second, in this classification both hybrids and non-hybrids clones are located at the same hierarchical level and hence, this nomenclature

lacks information in this regard. Additionally, no information about the maxicircle variant is given. Based on the drawbacks just mentioned, we propose to adopt the nomenclature shown in Fig 6: *T. cruzi* constitutes a complex composed of two main groups, 1 and 2, to differentiate them from I and II (since the latter lack correspondence with phylogenetic distances among clades), and four main lineages (clades): three belonging to group 1 (Aa, Bb, Db), and the fourth belonging to Group 2 (Cc); in addition, there are hybrid strains named BCb. Uppercase letters stand for nuclear genomes and lowercase letters indicate the maxicircle variant. This understanding and nomenclature can contribute to "put an order in the house", and focus the analysis of each clade to delve into their biological features. For example, in clade A we found: I) that at least two subgroups exist, one of them represented by Dm28c and Sylvio (Figs 5 and S9 and S10) and the second one by the so-called TcBat strains. Indeed TcBat strains used here clearly belong to clade A and carry the a-maxicircle (S9 Fig), therefore they all can be unambiguously classified as Aa. It is worth stressing that in this sense this new nomenclature avoids the temptation to propose more and more DTUs as new relatively non-divergent variants are discovered. Finally, to shed light on the origin of the close relationship between B and D, including the fact that they share the same maxicircle, whole genome analysis will help determine how hybridization and/or introgression events have occurred.

## Material and methods

### *T. cruzi* strains and DNA extraction

DNA was isolated by phenol extraction as described [51], and the integrity was checked by 1% agarose gel electrophoresis. The following strains were sequenced by Illumina technology: AP3-1, Colombiana, SC16, Sylvio X10 cl1 and Ort8-1 (TcI); ChaQ8-1, Cha_Q11-2 and ChaQ8-2 (TcI Bat); Y, PNM, Berenice, Esmeraldo cl3 and IVV cl4 (TcII); MT3663, Merejo do Anjico and 231 (TcIII); JoseJulio and AM64 (TcIV); BolFc10A (TcV); CLBrener (TcVI). The following strains were also sequenced by Nanopore: Y, MT3663, JoseJulio and BolFc10A. Finally, sequences from Dm28c, TCC, Bug21448, CLBrener Esmeraldo-like, CLBrener Non Esmeraldo-Like and *Tc. marinkellei* were retrieved from TriTrypDB and NCBI databases.

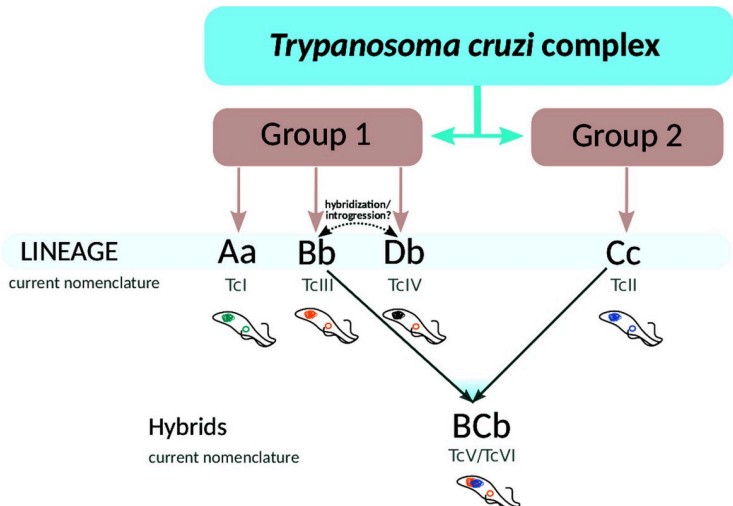

**Fig 6. Evolutionary relationships in the *Trypanosoma cruzi* complex.** *T. cruzi* is composed by two main groups 1 and 2, which does not correspond to the original I and II groups. Group 1 is divided in clades A, B and D, and group 2 contains clade C. Lower cases indicate the maxicircle variant (a, b, and c), and BC refers to hybrid strains.

## Discrete Typing Unit (DTU) determination

For DTU typing the following PCR products were amplified and sequenced as described in [19]: the intergenic region of spliced leader genes (SL-IR), the 24Sα subunit ribosomal DNA (rDNA 24Sα) and the A10 fragment. Size determination of PCR products was done onto 5% MetaPhor agarose gels.

## Library construction and sequencing

PacBio Sequencing was performed in the sequencing service of City of Hope (USA) using 10 µg of *T. cruzi* Y strain. Nanopore Sequencing was performed in our laboratory. Briefly, genomic DNA was fragmented to 20 kb using g-Tubes (Covaris, USA), according to manufacturer instructions and libraries were prepared with the kitEXP-NBD103/SQK-LSK108 (Nanopore, England) according to [52], starting from 1 µg of total fragmented genomic DNA. Libraries were run for 20 hours in R9.4 FlowCells (FLO-MIN106, Nanopore, England). Whole-genome Illumina sequencing libraries were performed as previously described in Pita el at. [53] using Nextera XT (Illumina). Paired-end reads were sequenced on the MiSeq platform (2 x 150 cycles).

## Assembly and annotation of maxicircle genomes

Long reads: FAST5 reads containing raw Nanopore signal were basecalled in real time using MinKNOW Nanopore software, and locally using Guppy toolkit (Oxford Nanopore Technologies). Porechop (https://github.com/rrwick/) was used to demultiplexing reads.

PacBio reads were assembled with HGAP v3 as described in [54] and Nanopore reads were assembled with Canu v1.8 [55]. Afterwards, backmapping Illumina reads with bwa [56], and using samtools for sam manipulation [57] and Pilon [58] were employed to polish the assembly. Contigs containing maxicircle sequences were recovered from the assembly using Blast [59] with previous *T. cruzi* maxicircle assemblies [6] used as subject.

Illumina reads: Reads belonging to kDNA were identified aligning all reads to already available kDNAs using BWA mem [56] with default parameters, and extracted with samtool [31] and bedtools [60]. Extracted maxicircle's reads were assembled using SPAdes version 3.8.0 [61] with default parameters. Scaffolds containing maxicircle sequences corresponding to the coding region were recovered from assembly using Blast [59] and controlled by base composition.

Annotation and Data handling: Maxicircles annotation was performed manually using Blast [59]. Coverage analyses were performed mapping illumina and long reads with BWA [56] and Minimap2 [62], respectively, and obtaining the amount of mapped reads using mpileup samtools [57]. R version 4.0.2 [63] with seqinr package, were used to obtain GC, GC and AT skews, and coverage plots. IGV [64] was used for alignment visualizations. Comparative analyses and visualization of coding regions were performed using ACT [65]. Circos [66] was used to create circular plots. Dotplots were performed using YASS web server [67].

## kDNA genes phylogenetic analysis

kDNAs obtained by illumina sequencing were analyzed in order to present at least 14 coding genes. A total of 17 strains fulfilled this requirement and were further processed. The entire coding region of the kDNAs were aligned using MAFFT v7.471 [68] with the *linsi* method and visualized with JalView [69]. The alignment was trimmed using trimAl [70] with option -gt 0.8. ML tree was generated by IQ-TREE [71] using GTR+F+G4 including 1000 bootstrap pseudoreplicates and visualized with FigTree (http://tree.bio.ed.ac.uk/software/figtree/).

## Nuclear vs maxicircle genes phylogenetic analysis

Single copy genes were retrieved using a pipeline from Pita *et al* (in preparation). Briefly, a clustarization using MCL software [72] is performed on all annotated genes in the Dm28c strain genome [54] to get rid of all highly abundant gene families. Then each survivor is used as a blastN query on several strains genomes assemblies -*T. cruzi* Dm28c, Sylvio, Berenice, Y, 231, MT3663, Jose Julio, AM64, TcBolFc10A, TCC, CL Brener Esmeraldo and Non-Esmeraldo-like haplotypes and *T. c. marinkellei* B7. The search was restricted to those genes which present only one HSP with 90% of query coverage (-qcovhsp) in all genomes. A gap penalty was setted to avoid genes with deletions (-gapopen 3 -gapextend 2). Each dataset was aligned separately using MAFFT v7.310 [68], and then concatenated using bash scripts. Same strains were used to perform a ML phylogenetic tree using kDNA, with the addition of the Esmeraldo strain, to compare with the CLBrener Esmeraldo-like haplotype. For both datasets Maximum Likelihood (ML) tree was generated using RAxML [73], with a partition scheme taking each gene independently. Since the substitution model test for each gene runed separately indicated that HKY plus gamma distribution was the best fitted most times, two RAxML runs were performed, one using GTRGAMMA and other using GTRGAMMA—HKY85. ModelGenerator v0.85 [74] software was employed to determine the best fitted model. The starting tree was found as the best-scoring ML tree using 20 randomized stepwise addition parsimony search (-p command). One hundred bootstrap pseudoreplicates were made (-b command) and then mapped onto the single most likely held tree topology (-f b command). In addition, to compare several approaches, ML trees were performed on both concatenated datasets with IQ-TREE [71] and PhyML [75] softwares. The phylogenetic trees were visualized and edited using the R package ggtree [76].

## Supporting information

**S1 Fig. DTU determination by multilocus PCR.**
(TIFF)

**S2 Fig. Base composition of maxicircles.**
(TIF)

**S3 Fig. Base composition analysis of maxicircle coding regions.**
(TIF)

**S4 Fig. Dot plot comparisons of all maxicircle genomes.**
(TIF)

**S5 Fig. Dot plot maxicircle visualization in other trypanosomatids.**
(TIF)

**S6 Fig. Sequencing coverage plots.**
(TIF)

**S7 Fig. Deletions in coding regions of C strains.**
(TIF)

**S8 Fig. Minicircle insertion in *T. cruzi* TcBolFc10A.**
(TIF)

**S9 Fig. Phylogenetic tree of maxicircle coding regions.**
(TIFF)

**S10 Fig. *T. cruzi* nuclear phylogeny.**
(TIFF)

**S1 Table. Short repeat conserved region.**
(XLSX)

**S2 Table. *T. cruzi* strains sequenced.**
(DOCX)

**S1 Data. List of 85 unique genes used to perform the nuclear phylogenetic maximum likelihood tree.**
(DOCX)

**S2 Data. List of 1023 unique genes used to perform the nuclear phylogenetic maximum likelihood tree without hybrid strains.**
(XLSX)

## Acknowledgments

We thank Dr Miriam Postan (Instituto Nacional de Parasitología "Dr. Mario Fatala Chabén", Buenos Aires, Argentina) for kindly giving us the TcBolFc10A strain.

## Author Contributions

**Conceptualization:** Luisa Berná, Gonzalo Greif, Carlos Robello.

**Data curation:** Sebastián Pita, Florencia Díaz-Viraqué.

**Funding acquisition:** Carlos Robello.

**Investigation:** Luisa Berná, Gonzalo Greif, Sebastián Pita, Fernando Alvarez-Valin, Carlos Robello.

**Methodology:** Gonzalo Greif, Sebastián Pita, Paula Faral-Tello, Florencia Díaz-Viraqué, Rita De Cássia Moreira De Souza, Gustavo Adolfo Vallejo.

**Resources:** Rita De Cássia Moreira De Souza, Gustavo Adolfo Vallejo.

**Supervision:** Fernando Alvarez-Valin, Carlos Robello.

**Validation:** Fernando Alvarez-Valin.

**Writing – original draft:** Luisa Berná, Carlos Robello.

**Writing – review & editing:** Carlos Robello.

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
