## [Decision Letter · Decision Letter 0]

27 Jul 2021

Dear Dr. Robello,

Thank you very much for submitting your manuscript "Maxicircle architecture and evolutionary insights into Trypanosoma cruzi complex" for consideration at PLOS Neglected Tropical Diseases. As with all papers reviewed by the journal, your manuscript was reviewed by members of the editorial board and by several independent reviewers. The reviewers appreciated the attention to an important topic. Based on the reviews, we are likely to accept this manuscript for publication, providing that you modify the manuscript according to the review recommendations. 

Sincerely,

Andrew Paul Jackson, Ph.D.

Associate Editor

Nilson Zanchin

Deputy Editor

Reviewer's Responses to Questions

**Key Review Criteria Required for Acceptance?**

**Methods**

-Are the objectives of the study clearly articulated with a clear testable hypothesis stated?

-Is the study design appropriate to address the stated objectives?

-Is the population clearly described and appropriate for the hypothesis being tested?

-Is the sample size sufficient to ensure adequate power to address the hypothesis being tested?

-Were correct statistical analysis used to support conclusions?

-Are there concerns about ethical or regulatory requirements being met?

Reviewer #1: see below

Reviewer #2: the objectives are clear and the methodology adopted is adequate

Reviewer #3: Very clear explanation of the purpose of the study and approaches adopted.

**Results**

-Does the analysis presented match the analysis plan?

-Are the results clearly and completely presented?

-Are the figures (Tables, Images) of sufficient quality for clarity?

Reviewer #1: see below

Reviewer #2: the results are presented in well-structured figures that adequately show the data obtained.

Reviewer #3: Very nice main figures, supported by comprehensive supplementary data.

**Conclusions**

-Are the conclusions supported by the data presented?

-Are the limitations of analysis clearly described?

-Do the authors discuss how these data can be helpful to advance our understanding of the topic under study?

-Is public health relevance addressed?

Reviewer #1: see below

Reviewer #2: I believe the discussion is the article's weakness. The data obtained do not contradict the data available in the literature about the nomenclature of T. cruzi strains. There are also extrapolations about hybridization events that are not adequately supported by the data. Other problems and suggestions are presented in the Summary and General Comments section

Reviewer #3: Very nice discussion, which accurately and fairly summarises the new data and provides context for the findings.

**Editorial and Data Presentation Modifications?**

Reviewer #1: see below

Reviewer #2: in figure 5 the TcV strain is incorrectly named as TcVI in nuclear phylogeny.

Reviewer #3: No changes needed.

**Summary and General Comments**

Reviewer #1: The authors present a very nicely executed study of T. cruzi maxicircle sequence assembly and diversity alongside a more limited analysis of nuclear sequences. The long read sequencing, corrected via illumine short reads, offers the most complete insight of maxicircle structure yet attempted. Furthermore, key findings – a large coding indel in the ND5 gene in two strains, and a apparently complete (or near complete) minicircle encoded in the maxicircle sequence – pose exiting questions, the former in relation to phenotypic differences between strains and the latter maybe shedding light on the origin minicircles themselves.

I must say I found the other focus of the paper, a re-exploration of T. cruzi intraspecific nomenclature, somewhat less convincing – in fact this element rather dilutes the interesting findings above. I was party to the discussion around the 2012 re-assigning of T. cruzi nomenclature. The resultant TcI – TcVI lineage assignment was as much about politics of the time as it was about biology. Such is the nature of compromise. I’m not sure how useful re-visiting the debate might be. That said, I do not object to the inclusion on the debate in the paper. 

I do, however, worry about the claim that TcV and TcVI are the progeny of the same cross and/or identical. If you look at multiple TcV and TcVI strains, fixed genetic differences between these clades, and fixed similarities within point clearly to separate origins (see /10.1371/journal.pntd.0001363 for example). As such I suggest the authors reserve judgement on this until multiple strains from each (V/VI) have been sequenced. I also think the authors should be careful about dismissing the possibility if maxicircle ingression without nuclear mixis – the authors present no data to the contrary. The clearest instances are the introgression of TcIV (b) maxicircles into TcI (e.g. PMC3323513) where there is no evidence of hybrid nuclear genomes split between divergent nuclear clades. 

Overall the study is well written, with some checking of English necessary. The tone is occasionally a little pugnacious – but there is no harm in that I suppose. 

Minor comments

Numbered lines would be very helpful in future.

Typo ‘Tse-Tse’ – Tsetse 

Introduction – the sentences on vectors on the bottom of page three needs re-drafting – they are poorly wriiten

Good use of the term 'dark matter'

Results

Can you clarify what is going on with Bug2148 - this seems to be a labelling error / laboratory mix up ? 

Did you detect any evidence for variation among maxicircles from the same clone

Surprised by how few single copy genes you could find orthologues between the Tc genomes - 85 unique conserved ?

Top of page 11 - do you mean 40 KDa or 40 Kbp ? Also - could you change the wording of 'To those authors our recognition.' – sounds a bit odd. 

Again - please be careful of sounding too dismissive - 'overvaluation of modern techniques' a little strong. Perhaps shift focus to emphasis the validity of more traditional approaches 

'irruption' – is not a word

Check where the acronyms CR and SR appear - they are somewhat confusing – make sure the appear where they are first used

‘Westemberg’ typo – is it not Westenberger ?

Reviewer #2: The manuscript is very important due to the assembly of the mitochondrial genome of different strains of T. cruzi. The results are interesting and the description of the structure of the mitochondrial genome in four regions is one of the most important data in the article.

 I do not agree with the final part of the discussion of the article, because all data presented agree with the separation of T. cruzi strains into DTUs-I to VI. The article does not present any evidence that only one hybridization event occurred, and the phylogenetic tree presented suggests two hybridization events by the position in the tree of the TcV and TcVI strains. It would be much more relevant for the manuscript to argue that the data obtained support the hypothesis of 4 ancestors, and it is contrary to the hypothesis that TCIII is the result of the hybridization event between TCI and TCII. Furthermore, it would be interesting to discuss why type b mitochondria prevails in all hybridization events and it is present in TcIV, can the size of the AT region favor the replication of this type of mitochondria?

Reviewer #3: In this manuscript, Berná and colleagues apply long-read, mainly Nanopore, DNA sequencing to examine the sequence composition of one of the two circles (the maxicircle) that make up the highly unusual, catenated DNA network of the Trypanosoma cruzi mitochondrial genome, which is called the kinetoplast. To do so, the authors compare maxicircle sequence from a number of parasite strains that span six of the discrete typing units (DTUs) of the T. cruzi species complex. Using these data, they reveal two aspects of T. cruzi biology. First, they reveal that most of the maxicircle genome is composed of 3 different types of repetitive sequence; thus, they reveal the ‘dark matter’ within the genome. Second, they use the newly assembled genomes of the DTUs to revisit the classification of strains and clades within the T. cruzi complex, and suggest a simplification is needed. 

In the interests of full disclosure, I confess that I was asked to review an earlier version of this paper, which was submitted to another journal, where it was ultimately rejected due to two main issues: lack of information around the potential functions of the new sequence elements revealed by long-read sequencing; and an over-assertive demand that T. cruzi DTU conventions be overturned by this work. This updated version of the paper is much improved, and both of these concerns have been admirably and comprehensively dealt with.

As such, I consider this submission to be very valuable: the sequencing efforts and results are comprehensive and will provide a rich data source for future work, and I now feel that the paper could initiate a useful discussion about the nomenclature used in the T. cruzi complex (indeed, the discussion is very fair and balanced in this regard). On this basis, I am content that the paper be accepted without (further) revision.

PLOS authors have the option to publish the peer review history of their article (what does this mean?). If published, this will include your full peer review and any attached files.

Reviewer #1: No

Reviewer #2: No

Reviewer #3: No

Figure Files:

Data Requirements:

Reproducibility:

References

---

## [Editor Report · Decision Letter 1]

10 Aug 2021

Dear Dr. Robello,

We are pleased to inform you that your manuscript 'Maxicircle architecture and evolutionary insights into Trypanosoma cruzi complex' has been provisionally accepted for publication in PLOS Neglected Tropical Diseases.

Best regards,

Andrew Paul Jackson, Ph.D.

Associate Editor

Nilson Zanchin

Deputy Editor

---

## [Editor Report · Acceptance letter]

23 Aug 2021

Dear Dr. Robello,

We are delighted to inform you that your manuscript, "Maxicircle architecture and evolutionary insights into Trypanosoma cruzi complex," has been formally accepted for publication in PLOS Neglected Tropical Diseases.

Best regards,

Shaden Kamhawi

co-Editor-in-Chief

Paul Brindley

co-Editor-in-Chief
